# Pharmacological Treatment of Bipolar Depression: A Review of Observational Studies

**DOI:** 10.3390/ph16020182

**Published:** 2023-01-25

**Authors:** Frederike T. Fellendorf, Edoardo Caboni, Pasquale Paribello, Martina Pinna, Ernesto D’Aloja, Sara Carucci, Federica Pinna, Eva Z. Reininghaus, Bernardo Carpiniello, Mirko Manchia

**Affiliations:** 1Department of Psychiatry and Psychotherapeutic Medicine, Medical University Graz, A-8036 Graz, Austria; 2Psychiatrische Universitätsklinik Zürich Zaft, University of Zürich, CH-8006 Zürich, Switzerland; 3Unit of Clinical Psychiatry, University Hospital Agency of Cagliari, 09124 Cagliari, Italy; 4Unit of Psychiatry, Department of Medical Science and Public Health, University of Cagliari, 09124 Cagliari, Italy; 5Forensic Psychiatry Unit, Health Agency of Cagliari, 09124 Cagliari, Italy; 6Section of Legal Medicine, Department of Medical Sciences and Public Health, University of Cagliari, 09124 Cagliari, Italy; 7Department of Biomedical Sciences, Section of Neuroscience and Clinical Pharmacology, University of Cagliari, 09124 Cagliari, Italy; 8Child and Adolescent Neuropsychiatry Unit, Department of Biomedical Sciences, “A. Cao” Paediatric Hospital, 09121 Cagliari, Italy; 9Department of Pharmacology, Dalhousie University, Halifax, NS B3H 4R2, Canada

**Keywords:** bipolar disorder, depressive episode, pharmacological treatment, systematic review, observational studies

## Abstract

The persistence of depressive morbidity is frequent in bipolar disorder, and the pharmacological management of this symptomatology often lacks effectiveness. This systematic review aimed to summarize the results of the naturalistic observational studies on the pharmacological treatment of bipolar depression published through April 2022. The certainty of evidence was evaluated according to the GRADE approach. In sum, 16 studies on anticonvulsants, 20 on atypical antipsychotics, 2 on lithium, 28 on antidepressants, and 9 on other compounds were found. Lamotrigine, quetiapine, aripiprazole, and ketamine were the most investigated compounds. Overall, the results support the recommendations regarding the effectiveness of lamotrigine and quetiapine. In contrast to the current recommendations, aripiprazole was shown to be effective and generally well tolerated. Additionally, SSRIs were shown to be effective, but, since they were associated with a possibly higher switch risk, they should be used as an adjunctive therapy to mood stabilizers. Lithium was only studied in two trials but was shown to be effective, although the serum concentrations levels were not associated with clinical response. Finally, ketamine showed divergent response rates with a low certainty of evidence and, so far, unclear long-term effects. Heterogeneity in diagnosis, sample sizes, study designs, reporting of bias, and side effects limited the possibility of a head-to-head comparison.

## 1. Introduction

Bipolar disorder (BD) is a chronic and highly disabling psychiatric illness affecting about 1% of the general population worldwide [1]. The clinical course of BD is typically characterized by alternating episodes of a depressed or elevated mood with intervals of well-being [2]. Longitudinal studies show that the onset of BD is frequently depressive polarity [3] and that the long-term affective morbidity, even in treated patients, is mainly depressive [4,5]. Indeed, BD patients experience substantial residual affective morbidity in 40–50% of weeks at follow-up, and about three-quarters are depressive [4,6,7]. These rates appear to be similar in each BD diagnostic subgroup (BD type 1 (BD1) = BD type 2 (BD2)) [6,8], although there is evidence of a predominance of depressive morbidity in BD2 compared to BD1 (BD2 > BD1) [9,10]. In a prospective study in which mood was assessed weekly with text-messaged ratings [11], conversely, individuals with BD1 reported more days of depression and were less likely to improve with time than BD2 patients.

The persistence of depressive morbidity, despite pharmacological treatment, indicates that achieving an adequate response for these mood states remains a challenge in BD [12]. A systematic review and meta-analysis of randomized controlled trials (RCT) found that only the glutamate N-methyl-D-aspartate (NMDA)-receptor antagonist ketamine and, to a lesser extent, the dopamine-receptor agonist pramipexole were superior to a placebo in treating bipolar depression [12]. However, ketamine effectiveness is mainly in the short term (up to 24 h), and there is scant knowledge on its impact on the long-term remission of bipolar depression [13].

In this context, it appears crucial to synthesize the available evidence coming from observational studies on the pharmacological treatment of depression for BD patients. Therefore, we performed a systematic review and a critical appraisal of the evidence using specific grading criteria. 

## 2. Results

### 2.1. Systematic Search

Of the 7486 records identified in Medline and the 9865 identified in Embase, 4170 were duplicates. Of the remaining studies, 73 publications met the inclusion criteria and were further investigated in detail. The search found 16 publications on anticonvulsants [14,15,16,17,18,19,20,21,22,23,24,25,26,27,28,29], 20 on atypical antipsychotics [21,30,31,32,33,34,35,36,37,38,39,40,41,42,43,44,45,46,47,48], 2 on lithium [49,50], 28 on antidepressants [51,52,53,54,55,56,57,58,59,60,61,62,63,64,65,66,67,68,69,70,71,72,73,74,75,76,77], 2 on CNS stimulants [78,79], 1 on pramipexole [80], 2 on hormones [81,82], 2 on NMDA antagonists [83,84], 1 on a peroxisome proliferator-activated receptors (PPAR)-γ agonist [85], and 1 on antibiotics [86]. The results are summarized in several tables and presented according to the pharmacological class of the drugs.

### 2.2. Anticonvulsants

Table 1 presents the descriptions of the studies conducted on anticonvulsants. Specifically, two studies evaluated the effect of zonisamide, one evaluated the effect of gabapentin, one evaluated the effect of divalproex/valproate, one evaluated the effect of carbamazepine, one evaluated the effect of topiramate, one evaluated the effect of lamotrigine or quetiapine, and nine evaluated the effect of lamotrigine. Zonisamide was shown to be effective in eight-week settings according to the Montgomery Asberg Depression Rating Scale (MADRS; [87]), the mania rating scale from the “Schedule for Affective Disorders and Schizophrenia” (SADS; [88]) [14], the Clinical Global Impression Bipolar Depression (CGI-BP-D; [89]), and the Inventory of Depression Symptomatology (IDS; [90]) [15]; however, it is worth noting that the certainty of the evidence was “very low”, due to a high discontinuation rate of 50% [14], a low depression rate (21/60), and a high discontinuation rate [15]. An open-label trial by Wang et al. with a “low” certainty of evidence grading found a 53% decrease in depressive symptoms measured by the Hamilton Depression Rating Scale (HAMD; [91]) with a gabapentin treatment [16]. One patient (equal to 5%) suspended the treatment for cognitive impairment. Only one trial about valproic acid found a statistically significant reduction in the scores of MADRS, HAMD, the Young Mania Rating Scale (YMRS; [92]), and CGI in a sample of 28 BD patients [17]. The study design and conduction achieved a “low” certainty of evidence. The only study about 21-day carbamazepine administration determined a statistically significant reduction in HAMD scores, 63% in 36 BD depressed patients, representing a “low” certainty of evidence [18]. McIntyre et al. investigated the effect of topiramate as an add-on to other mood-stabilizing treatments for 56 participants who were depressed at the time of starting, 42 who were in a mixed state, 8 who were in a hypomanic state, and 3 who were in a manic state. The depressive symptoms measured by MADRS and CGI improved significantly after two weeks and again from week 2 to week 16 [19]. Two chart reviews and eight prospective studies concerned lamotrigine. A chart review by Montes et al. found a statistically significant global reduction in the CGI-BP-D of lamotrigine either as monotherapy or an add-on treatment [20]. Quoting the authors, a limitation consisted of the fact that “the independent effect of lamotrigine could not be determined since it was used in many cases as adjunctive therapy, without taking into account possible synergistic effects of other agents”. Furthermore, the cohort consisted of heterogenous patients with a cyclothymic disorder, or a disorder not otherwise specified (NOS), with frequent comorbid psychiatric disorder, e.g., personality disorder, and participants in all affective states were included. Therefore, grading yielded a “very low” certainty of evidence. Another retrospective chart review in 31 BD2 patients with treatment-resistant depression (TRD), followed for an average of 19.4 months, showed that lamotrigine (daily dose between 50 and 400 mg) alone or as an add-on treatment was well-tolerated and effective: using the CGI-Improvement scale (CGI-I), 52% were judged as “very much improved”, 32% as “much improved”, and 16% as “minimally improved” [28].

A prospective naturalistic study by Ahn et al. showed that either adding lamotrigine to an inadequate response to quetiapine monotherapy or adding quetiapine to an inadequate response to lamotrigine was effective in reducing depressive symptoms, as shown by reductions in the CGI-Severity (CGI-S) and CGI-Functioning (CGI-F) scales [21]. Improvements were even higher when quetiapine was the add-on drug. The combination treatment was quite safe. The other seven prospective studies on lamotrigine as an add-on to a formerly insufficient treatment were concordant in showing a statistically significant improvement in depressive symptoms (Table 1) [22,23,24,25,26,27,29]. No difference in the response rate was found between participants with unipolar depression and BD [24]. Chang et al. found, in 109 bipolar depressed individuals, an association with a poorer response to lamotrigine, a higher number of hospitalizations, and a history of suicide attempts [22]. Four of the studies had a “very low” and four a “low” certainty of evidence, while the downgrading reasons were a low optimal information size with effects on low imprecision.

For a detailed description of the side effects of anticonvulsants, see Table 1. In sum, switching to (hypo)manias was not reported often. Regarding side effects, the two studies on zonisamide had a high dropout rate because of the development of nausea, vomiting, and sedation [14,15]. The only study on gabapentin reported mild sedation and weight gain [16]. Additionally, valproate was associated with weight gain [17], while topiramate was associated with weight loss and a reduction in tremors [19]. There was no early discontinuation of valproate due to side effects. Some cases of fever and rush were shown under carbamazepine treatment [18]. Most studies showed that lamotrigine was well-tolerated, with no serious side effects. The most common side effects were rashes, headaches, and sleep alterations. Additionally, one study reported a switch to (hypo)mania with an increase in YMRS; however, the authors did not report more detailed data [29]. Finally, Joe et al. compared a standard titration (*n* = 132) with a slower titration (*n* = 127) of lamotrigine and found significantly fewer rashes in the slow-titration group [23].

### 2.3. Second-Generation Antipsychotics

Our systematic search identified 20 studies (3 retrospective reviews, 1 comparison of RCTs, and 16 open prospective studies) that analyzed the effectiveness of antipsychotics in bipolar depression (see Table 2). Of them, one investigated olanzapine monotherapy, one investigated a combination treatment of risperidone and olanzapine, one investigated olanzapine and quetiapine, five investigated quetiapine monotherapy, seven investigated aripiprazole monotherapy, two investigated lurasidone monotherapy, one investigated ziprasidone monotherapy, one investigated risperidone (long-acting injectable (LAI), and one investigated brexpiprazole monotherapy.

Olanzapine monotherapy was associated with a response rate of 9/20 (with eight patients achieving symptom remission). Grading was a “low” certainty of evidence due to the open-label design [30]. Additionally, a statistically significant reduction in the HAMD score with risperidone as well as with olanzapine treatment was observed by McIntyre et al., with no differences between the two groups [31]. Four studies evaluated the effectiveness of quetiapine in bipolar depression. Specifically, Porcelli et al. found an improvement in depressive symptoms and a trend toward a better improvement with 600 mg/day instead of 300 mg/day of quetiapine extended release (ER) [34]. Dell’Osso et al. investigated switching from quetiapine immediate release (IR) to ER and found good efficacy, with a reduction in both the HAMD and the Hamilton Anxiety rating scale (HAMA; [93]) and no change in compliance [32]. A retrospective review by Shajahan et al. found a 69% improvement for all depressive subtypes, with the best improvement in BD mania, followed by BD depression [35]. Additionally, Suppes et al. showed in their chart review an improvement in patients receiving adjunctive quetiapine. Depressed and cycling participants who received a mid-level dose (100–399 mg) were most responsive between weeks seven and nine [36]. As mentioned previously, the study of Ahn et al. investigated the add-on of either lamotrigine or quetiapine and found both to be effective, but quetiapine was even more so [21]. Kishi et al. compared the effectiveness of one RCT investigating olanzapine and one investigating quetiapine ER and found no difference regarding response and remission rates [33]. Seven studies investigated the use of aripiprazole in bipolar depression: one with a retrospective design and six with a prospective design. Three studies (including a chart review) assessed the effectiveness of this second-generation antipsychotic as monotherapy, and all observed clinical improvement. Dunn et al. found an enhancement in MADRS with a better trend in the monotherapy group, though it was statistically not significant [37]. This result was also confirmed by Mazza et al. [41]. The same study group found an improvement during aripiprazole monotherapy, with a reduction in MADRS, no effect on YMRS, and a reduction in the Snaith–Hamilton Pleasure Scale (SHAPS) score, indicating a decrease in anhedonia (52% anhedonia at baseline and 20% at the end) [42]. McElroy et al. showed that response and remission rates were not affected by whether aripiprazole was received alone or as an add-on [43]. Finally, Malempati observed improvements in MADRS, CGI, and complete functional recovery on the Sheehan Disability Scale (SDS; [94]), when using aripiprazole as an adjunctive therapy [40]. Two studies on lurasidone were found. One, conducted by Ketter et al., analyzed a large cohort of 817 individuals, showing that lurasidone administered either as an adjunctive therapy or monotherapy was effective in a six-month follow-up setting [44]. Another study by Miller et al. investigated lurasidone as an adjunctive treatment and found the CGI-BP-S values decreased in depressed patients, while no statistically significant change was observed in subsyndromal depression [45]. Liebowitz et al. performed an open trial to test the clinical effectiveness of ziprasidone monotherapy in BD2 depressed patients [46]. The authors, assessing effectiveness with CGI-S, HAMD, HAMA, and MADRS, found significant improvement in CGI-S and HAMD beginning after one week and in HAMA and MADRS beginning after two weeks. 

Only one trial concerned LAI preparations [47]. Specifically, MacFadden et al. tested adjunctive LAI risperidone in controlling depressive symptoms in a sample of 162 BD patients. The authors found that 53.3% achieved remission, defined as YMRS total score ≤8, MADRS total score ≤ 10, and CGI-BP-S score ≤ 2. Finally, one study tested the efficacy of brexpiprazole [48]. Only patients with moderate to severe depressive symptoms (MADRS > 25) were included, to reduce the bias of spontaneous improvement. Quality of life and cognition were evaluated as secondary outcomes. No participants discontinued the trial because of side effects, and a significant reduction in MADRS and IDS-SR30 values was observed.

For the side effects reported during second-generation antipsychotics treatment, see Table 2. In general, they showed a good tolerability profile, as only a few studies reported early discontinuation, mostly due to insomnia, headaches, dizziness, akathisia and worsening of mood state (aripiprazole), weight gain, and dyslipidemia. Olanzapine treatment and quetiapine treatment were mainly associated with weight gain and somnolence, whereas Kishi et al. found in their comparison that olanzapine was more associated with weight gain and decreased levels of high-density cholesterol, while quetiapine ER was associated with a greater risk of somnolence [33]. Of the seven studies on aripiprazole, six showed akathisia [37,38,40,41,42,43], four showed nausea [37,38,39,43], and two showed tremors [38,43]. Regarding appetite, McElroy et al. showed increased and decreased appetite. A mild weight gain was reported by Malempati et al. [40], while Dunn et al. [37] demonstrated an increase in glucose levels, with concentrations in the normal range. All other trials did not find any weight change during aripiprazole treatment, but, of note, in most studies, the treatment duration was only of some weeks. Regarding lurasidone, one study showed that 6.9% (adjunctive treatment) and 9.0% (monotherapy) of the sample discontinued due to an adverse effect, mostly weight gain and dyslipidaemia [44]. The primary outcome of the other study on lurasidone was to estimate the discontinuation rate, which appeared as high as 77% (54.1% because of inadequate tolerability, 16.4% because of inadequate efficacy, and 6.6% because of other reasons). The most common side effects were akathisia, sedation, and weight gain [45]. Ziprasidone was associated with tremors, a low white blood cell count, and other less specific symptoms such as insomnia and headache [46]. The study of MacFadden et al. observed tremors and weight increase after LAI risperidone therapy [47]. The only study about brexipiprazole reported that akathisia was associated with the treatment [48].

### 2.4. Lithium

We found two observational studies that assessed the effectiveness of lithium as a treatment for bipolar depression. Goodwin et al. used a “15 points multi-item” scale to assess lithium effectiveness. Overall, 80% showed some improvement, while about one-third (12 out of 40 patients) showed an unequivocal response to lithium. In comparison to unipolar participants (4 out of 12), there was a statistically significant improvement in BD [49]. Machado-Vieira et al. investigated the effect of lithium monotherapy, using either a higher (≥0.5 mmol/L) or a lower (<0.5 mmol/L) dose in 29 individuals with BD1 and BD2 in a six-week study [50]. The results showed an improvement in depressive symptoms measured by HAMD and a remission rate of 62%. There was no difference in antidepressant efficacy between the two dose regime groups, but significantly more side effects, namely nausea, restlessness, headache, and cognitive complaints, were found in the higher dose group.

### 2.5. Antidepressants

Our systematic search identified four studies on the effectiveness of various antidepressants, one study comparing hypnotic drugs and sedative antidepressants, seven studies on a selective serotonin reuptake inhibitor (SSRI), one study comparing two serotonin-noradrenaline reuptake inhibitors (SNRI), one study on tricyclic antidepressants, two studies on agomelatine, and nine studies on ketamine. 

One study by Tundo et al. investigated the effectiveness of various antidepressants in 154 unipolar, 49 BD1, and 52 BD2 participants with a “low” certainty of evidence [51]. Using HAMD to assess the response rate, 75.5% of BD1 and 75% of BD2 patients achieved remission. A switch was observed in 2.9% of both groups, specifically in two patients with BD1 and in one patient with BD2. The other side effects and adverse effects were unreported, as was the discontinuation rate. Shvartzman et al. [52] focused on the rehospitalization rate of 98 BD patients over one year, with the adjunction of various antidepressants to mood stabilizers, finding a statistically significant lower readmission rate and a longer time to rehospitalization due to a depressive episode. However, because of the relatively short observation period, the results were downgraded to a “very low” certainty of evidence. A different outcome was observed by Hooshmand et al., with a “low” certainty of evidence, where 503 bipolar outpatients were followed prospectively for up to two years [53]. The prevalence of baseline antidepressant use was significantly lower among recovered versus depressed patients (31.4% versus 44.4% *p* = 0.04). Baseline antidepressant use was not significantly related with the time to recover. The authors found a hastened depressive recurrence using CGI-BP-S among recovered patients taking antidepressants; however, since the study had a naturalistic approach, they concluded that causality could not be assessed. Furthermore, the hastening of depressive recurrence seems not to persist if the patients with mood elevation episodes prior to depressive episode recurrence were censored, suggesting that this clinical observation may correlate with a global hastening of illness cycles in high-risk patients. Saiz-Ruiz et al. examined the use of either hypnotic drugs or sedative antidepressants in a group of 53 bipolar patients experiencing insomnia [54]. No standardized tests were used by the researchers nor were any side/adverse effects reported, indicating a “very low” certainty of evidence grading. The use of antidepressants was associated with a worse prognosis in this group, with the parameter of a “symptom-free period” being longer in people treated with hypnotics (18.81 months versus 12.73 months); however, the results were not statistically significant (*p* = 0.6).

Frankle et al. conducted a retrospective chart review investigating whether there was a relationship between the use of various antidepressant drugs (specifically SSRIs, bupropion, venlafaxine, monoamine oxidase inhibitors, mirtazapine, and psychostimulants) and the length of the major depressive episode in bipolar patients [55]. Side and adverse effects were not reported, and there was no evidence of any episode length shortening nor of any induced mood switch. The authors suggested that antidepressants might be neither useful nor markedly dangerous in the treatment of bipolar depression.

For the description of the studies investigating antidepressant monotherapy, see Table 3.

Of the studies on SSRIs, three investigated the effectiveness of fluoxetine [56,57,58], one investigated the effectiveness of add-on citalopram [61], one investigated the effectiveness of add-on escitalopram [60], one investigated the effectiveness of paroxetine [59], and one compared paroxetine and mianserin. All three studies on fluoxetine—with two reporting on an overlapping cohort—showed a statistically significant improvement in depressive symptoms [56,57,58]. One of them did not find a link between clinical features and drug plasma concentration [56]. Another study demonstrated a greater reduction in depression symptoms and a higher frequency of hypomanic episodes amongst rapid cyclers, with no difference in response rate or remission rate and no change in mean mania rating scores though [58]. Amsterdam et al. reported in the two publications on the same cohort that BD patients receiving fluoxetine as monotherapy fulfilled, in 6/148 cases, the criteria for hypomania and, in other 29/148, the criteria for subsyndromal hypomania. Additionally, one case of severe mania and one suicide attempt were found in the cohort. Kupfer et al. tested the effectiveness of citalopram and classified 64% as responders and 36% as non-responders [61]. None of the patients discontinued because of adverse effects. A study on escitalopram showed, besides a good response, four dropouts due to manic switch (*n* = 1), hypomanic symptoms (*n* = 2), and hospitalization because of psychosis and suicidal ideation (*n* = 1) [60]. Further, Baldassano et al. conducted a chart review on paroxetine in 20 bipolar depressed individuals, who had failed at least one standard antidepressant therapy in the past and found 85% failed to improve within 12 weeks from the start. At harvest, 65% showed much or very much improvement, and the Global Assessment of Functioning (GAF; [96]) mean improved significantly. While 10% switched into mania, they had a history of switching [59]. In another study, Mertens et al. investigated the difference between paroxetine and the tetracyclic substance mianserin and found a significant improvement according to HAMD but no difference between the groups. Nausea and headache were seen in four patients taking paroxetine, and somnolence was seen in six patients taking mianserin [97]. Only one study investigated the effectiveness of SNRIs, whereby duloxetine was found to be more effective than venlafaxine in treating depressive symptoms. Concerning tolerability, none of the patients had to stop treatment because of adverse effects [63]. One study by Kocsis et al. tested the response of the tricyclic antidepressants amitriptyline and imipramine for four weeks, after being drug-free for two weeks. They found an increase in HAMD and SADS. Side effects, e.g., switching, were not reported in the publication. The authors split the sample into psychotic depressed, moderately depressed, and severely depressed and found better outcomes in the non-psychotic depressed group, with no difference between bipolar and unipolar depressed participants [64].

One study on agomelatine classified 81% of the patients receiving agomelatine as adjunctive therapy to lithium or valproate as responders. Of note, the response rate was equal for 78.6% of BD patients treated with lithium and agomelatine and 57.1% who were treated with valproate and agomelatine. During the six-week period, no patients experienced an emergent adverse event leading to study discontinuation. During the optional period, three lithium-treated patients withdrew from the study (one for agitation, one for mania, and one for hypomania) [65]. In another study, 64% showed a response after six weeks, and 86% showed a response after 36 weeks. Moreover, 54.5% of individuals taking lithium responded after 6 weeks, 90.9% responded after 36 weeks, while among participants taking valproate 70.6% responded at 6 weeks and 82.4% at 36 weeks. Four patients treated with valproate and agomelatine reported pseudo-vertigo and hypomania, and two participants treated with lithium and agomelatine reported insomnia and mania and, therefore, dropped out at week six. Two more cases of hypomania were found at week 36 [66].

Ketamine IV for bipolar depression was investigated by 12 studies, all giving a “low” certainty of evidence (see Table 3). Most of them continued with mood stabilizers but stopped other antidepressants one to two weeks before ketamine treatment. Studies by Permoda-Osip et al. [71] and Rybakowsky et al. [72] showed that 50% of participants receiving single infusions achieved at least a 50% reduction in HAMD scores. A study by Ionescu et al. showed a statistically significant reduction in depressive symptoms (HAMD) in BD patients with anxious and non-anxious manifestations [67]. Even though there was no specific reference to the adverse effects in the study, it was indicated that they did not differ between the two groups. Furthermore, McIntyre et al. showed a significant reduction in depressive symptomatology after four ketamine infusions in TRD individuals (defined as an insufficient response to two drugs), either unipolar or bipolar, who were measured with a decrease in Quick Inventory for Depression Symptomatology-Self Report-16 (QUIDS-SR16; [98]), Generalized Anxiety Disorder-7 Scale (GAD-7), and SDS [68,69]. Moreover, a positive effect on anhedonia [75]; anxiety, overall psychosocial function, and suicidal ideation [68]; cognitive domains such as processing speed and verbal learning [77] as well as a moderating effect of total functional disability; and the subdomains social life and family life/home responsibilities [69] were found. No serious side effects and no exclusions because of side effects were observed, but the following symptoms were found during and after treatment: dizziness, drowsiness, confusion, depersonalization, derealization, blurred vision, double vision, nausea, and headache [68]. Another paper intending to measure the efficacy of repeated ketamine dosages in severely depressed TRD individuals found a response rate of 68.0% and a remission rate of 50.5% in the total sample and, in the bipolar subgroup, a response in 14/20 and remission in 6/20 [73]. 

Three studies on the short- versus long-term effects of intravenous ketamine were conducted. Pennybaker et al. assessed the data about a single shot of ketamine in 122 depressed individuals (bipolar or unipolar) by analysing the results of four different studies on ketamine. Overall, 32.5% of the participants had an antidepressive response after 24 h, and 12.9% had an antidepressive response after two weeks (only 93/122 participants were assessed at the two-week time point). The responders at week 2 had a greater response after 230 min and after 24 h than the two-week non-responders [70]. Zhuo et al. found, after 10 infusions every two days, a significant HAMD reduction after one week, but a relapse occurred during the second week and even more severely depressive symptomatology occurred by day 21 than at baseline. There was a drop out of 13 patients because of side effects, which were not reported in detail [74]. Li et al. found a 46.6% maintenance response after nine months and a 25% risk of relapse within two weeks after a 12 day ketamine treatment [76]. Finally, Lara et al. prescribed sublingual ketamine, at a lower dose than intravenous, in a total to 26 TRD subjects who were either bipolar or unipolar. They found a remission or clear response in mood and sleep in 77.0%, but they only used non standardised questions for quantification. The application was tolerated quite well, with only mild light-headedness reported, mostly for less than 30 min. No manic, psychotic, or dissociative symptoms were observed [95].

### 2.6. Other Pharmacological Agents

Two more studies about CNS stimulants other than ketamine were identified. Ketter et al. evaluated, with a “low” certainty of evidence, the effect of six-month armodafinil as adjunctive therapy to mood stabilizers in 506 BD1 patients. Add-on armodafinil was effective in reducing depressive symptoms, as shown by a significant reduction in the IDS-C30 and QIDS-C16 scores. Further, global functioning, measured with GAF, improved substantially. The response rate and safety showed it to be profitable, but 7% of the participants discontinued because of adverse effects (none because of akathisia or somnolence but one patient because of acute psychosis) [78]. In another study, Parker et al. determined the effectiveness of methylphenidate and dexamphetamine in 26 BD1 and 23 unipolar patients in a variable duration (mean 57 weeks). The authors observed that 34% showed a significant clinical improvement, and 30% showed a partial improvement, while no improvement was found in 36%. Of note, no standardized definition of clinical response was provided in this study. There was no mention of discontinuation because of adverse effects [79].

One trial included 37 individuals with BD and 79 with major depression, with a failure of response to two antidepressants, and tested the efficacy of the dopamine agonist pramipexol as an add-on for 24 weeks. A 74.1% response rate and a 66.4% remission rate, measured with a HAMD reduction, were found. Eight participants dropped out due to side effects, and one showed hypomanic symptoms [80]. Two trials investigated hormones as an additional approach in the treatment of bipolar depression. One study with a “low” certainty of evidence showed the effectiveness of triiodothyronine augmentation in 159 patients with either BD2 or BD NOS diagnoses suffering from depression, using a retrospective approach. The authors found that 84% of BD2 and 85% of BD NOS patients improved (as shown by a statistically significant reduction in CGI and GAF, of which 38% and 32%, respectively, were considered in remission) [81]. Triiodothyronine was well-tolerated overall, with tremors (responding to dose reduction), osteoporosis (even though the bone loss was not systematically assessed), atrial fibrillation, and, generally, hyperthyroidism being the most common side effects. The discontinuation rate was equal to 10% of patients (only one out of three discontinuing patients was defined as being caused by the patient’s concern for bone loss). Amsterdam et al. investigated the intramuscular injection of a gonadotropin-releasing hormone versus a placebo in 30 individuals, who were either bipolar or unipolar depressed (with a “very low” certainty of evidence). They did not observe any antidepressive effects with one administration or in a three-day follow-up [82]. Further, one study investigated the effectiveness of memantine, a weak non-competitive antagonist at the NMDA receptor, and one other study analyzed dextromethorphan, a cough suppressant (acting as an NMDA antagonist, sigma receptor agonist, mu Opioid receptor agonist, SERT blocker, and NET blocker). Memantine was studied by Serra et al. in a naturalistic open study with a “low” certainty of evidence for at least three years, including BD1 and BD2 patients. An improvement in CGI-BP was found after three years of treatment in comparison to the stagnation found during the three years before, especially in rapid cycling patients [83]. Dextromethorphan, as an additional treatment to existing drugs, was studied by Kelly et al. in a retrospective chart review. A significant improvement was detected in CGI-I scores after 90 days, with 25% of patients discontinuing treatment because of adverse effects, chiefly nausea (typically for the action on opioid receptors) [84]. We identified one open-label study on a PPAR-γ agonist with a “low” certainty of evidence [85]. In a cohort of 34 BD patients, a statistically significant improvement following eight weeks of pioglitazone treatment emerged in clinician and self-reported assessments of depression (IDS-C30 and QUIDS) and anxiety (HAMA) as well as in functional improvement (SDS). Only 6% of patients discontinued treatment because of activation/irritability. Other reported side effects were dizziness, irritability, increased appetite, and peripheral oedema [85]. One study considered the neuroprotective effect of the antibiotic minocycline, as mediated by its antioxidant properties, in 20 patients with BD. The aim was both to evaluate the efficacy of an 8-week treatment and to test minocycline’s relationship with glutathione levels (measured with proton magnetic resonance spectroscopy). Minocycline was titrated up to 300 mg/die, and safety and efficacy assessments were completed every two weeks. A large decrease in depressive symptoms (MADRS, CGI-S, and CGI-BP) was observed; response and remission rates achieved 50% and 40% respectively; and improvements in daily functioning and quality of life (Quality of Life, Enjoyment, and Satisfaction Questionnaire (Q-LES-Q; [99]); Longitudinal Interval Follow-up Evaluation—Range of Impaired Functioning Tool (LIFE-RIFT; [100])) were measured. Unexpectedly, though, the authors observed a paradoxically higher glutathione increase in non-responders than in responders. Moreover, cognitive performances appeared unaffected by minocycline [86].

## 3. Discussion

This review aimed to synthesize the knowledge about the pharmacological treatment of bipolar depression in real-world settings. Therefore, the results of several naturalistic observational studies were reported in detail. In sum, 16 studies on anticonvulsants, 20 studies on atypical antipsychotics, 2 studies on lithium, 28 studies on antidepressants, and 9 studies on other compounds were found. Of interest, lamotrigine, quetiapine, aripiprazole, and ketamine were the most frequently investigated compounds. Most of the studies reported the partial or complete resolution of depressive symptomatology, but heterogeneity in diagnosis (BD1, BD2, NOS, and unipolar depression), sample sizes, study designs, reporting of bias, and side effects limited the possibility of a head-to-head comparison. The validity of the evidence was evaluated according to the GRADE guidelines. However, the naturalistic design of the included studies impacted the quality assessment, with most studies having a “low” certainty of evidence grading. Nonetheless, the findings of this qualitative data synthesis can elucidate the effect of classes of drugs as well as single compounds on depressive symptomatology in BD. Overall, this review supports the recommendations of the 2018 updated guideline of the Canadian Network for Mood and Anxiety Treatments (CANMAT) and the International Society for Bipolar Disorders, based on large RCTs. They reported, for acute bipolar depression, level 1 evidence for quetiapine, lurasidone plus lithium or valproic acid, SSRIs, bupropion, and cariprazine [2]. Level 2 recommendations were achieved for lithium alone, lamotrigine as monotreatment and add-on treatment, lurasidone alone, valproic acid, and olanzapine plus fluoxetine [2]. Lithium, quetiapine, and aripiprazole were each recommended as a first-line long-term maintenance treatment monotherapy [101]. 

All investigated anticonvulsants (zonisamide, gabapentin, valproic acid, carbamazepine, and lamotrigine) showed improvement in the depressive symptomatology in depressed subjects with BD. Although it is not possible to perform a head-to-head comparison between the different anticonvulsants, our findings showed that a larger proportion of study participants dropped out during trials with zonisamide, mainly due to the onset of severe side effects. Additionally, the certainty of evidence for the studies on zonisamide was “very low”. As the sample sizes in the trials investigating gabapentin, valproic acid, carbamazepine, and topiramate were rather small, and only one trial was conducted on each compound, the implications for clinical practice remain unclear. Of note, a review of RCTs by Vazquez et al. found a good effectiveness by anticonvulsants in general, although carbamazepine and valproate were not well-tolerated [102]. All the studies included in this review showed an improvement in depressive symptomatology, mostly within 12 weeks. Furthermore, tolerability was acceptable with only mild side effects and an absence of increased risk for the manic switch. 

Lamotrigine was investigated in 10 studies (five with “low” and five with “very low” certainty of evidence gradings), allowing for more derivations. The CANMAT group reported differing results for RCTs on lamotrigine effectiveness, showing either a lack of efficacy [103] or good efficacy [104]. The authors highlighted the short duration of the trials and the relatively small doses administered as the main limitations of the available evidence. Other studies provided evidence of the effectiveness of lamotrigine for bipolar depression. A decreased occurrence of skin rashes was found with a slower titration compared to a normal titration for lamotrigine [23]. Only one trial evaluated the difference between BD and unipolar patients and found no differences in the response [24]. One study showed a greater response in patients with diagnoses within the BD spectrum than in BD itself [20]. Furthermore, the addition of lamotrigine to lithium was more effective than the addition of a placebo [105], which indicates the clinical relevance of this combination treatment.

The most investigated antipsychotic was aripiprazole, followed by quetiapine. Only two studies compared two antipsychotics, namely olanzapine/risperidone and olanzapine/quetiapine, finding no difference in outcome [31,33]. All trials studying aripiprazole, quetiapine, ziprasidone, olanzapine, and risperidone found an improvement in depressive symptoms. These studies showed that atypical antipsychotics were effective in reducing depressive symptoms either in monotherapy or as an additional treatment. Further, most of these studies observed positive effects on global functioning. Another recently marketed second-generation antipsychotic, lurasidone, had relatively high rates of discontinuation due to the onset of side effects [44]. The selected findings on quetiapine suggest that the dosage regimen might play a role in effectiveness, with one study reporting a superior effect for 600 mg [34] and another reporting a superior effect for 100–400 mg [36]. The CANMAT guidelines recommend quetiapine and lurasidone as treatment options for bipolar depression. However, as seven studies in this review showed a good response rate and, except for akathisia, good tolerability of monotherapy and an add-on therapy with aripiprazole, our findings highlight a potential clinical utility of this second-generation antipsychotic for the management of bipolar depression.

According to CANMAT, RCTs on lithium efficacy on bipolar depression are conflicting [2,103] and indicate that higher serum levels might be needed for antidepressant efficacy (0.8–1.2 mmol/L). Of note, a study by Machado-Vieira found an improvement in depressive symptoms but no difference in plasma levels (≥0.5 versus <0.5 mmol/L) [50]. A seminal study included in this review found a significant better response in individuals with BD than in those with unipolar depression [49]. Although lithium is known to be effective in preventing bipolar episodes, evidence on acute depression is scant and demands more research.

Our findings, that adjunctive antidepressants in general and SSRI in particular are effective on depressive improvement, are in line with a meta-analysis of RCTs [106]. However, almost any trial investigating SSRIs reported the occurrence of at least one (hypo)mania. It is worth noting that the relation of this mood switch to the antidepressant is not always clear. In general, antidepressants, even SSRIs with a relatively low risk for switching, must be used carefully, possibly as an add-on therapy to an adequately dosed mood stabilizer. As agomelatine was demonstrated to be effective as an adjunctive therapy, it should be further evaluated in an adequately powered RCT.

A recent focus of this area of research is to investigate the short- and long-term effects of ketamine. A recent review of RCTs found a reduction in depressive symptoms 24 h after infusion, but no further efficacy in the longer term [107]. Nine trials with a “low” certainty of evidence were included in this review, as they focused only on or also on BD patients. The response rates showed a large range, varying from 27% to 77%. This might be due to different measurement times because of the rapid but only temporary antidepressant effect of ketamine [107]. One included study found a better response after 24 h than after two weeks, but those who responded better after two weeks also had a good response in the short term [70]. Of note, another study demonstrated a short response but with a relapse with more depressive symptomatology than at baseline [74]. Most studies did not find or report any differences in the response between BD and the unipolar depressed. Additionally, the side effects were not constantly reported, making it difficult to infer information for clinical decision making.

Finally, other compounds demonstrated some degree of efficacy in treating bipolar depression. The observed improvement with armodafinil could not be demonstrated by RCTs [108]. Thyroid hormones have been widely used to treat depression, especially as an add-on [109]. Therefore, these findings support the data about the efficacy and good tolerability of this treatment option [81]. 

To conclude, the findings of this review attempt to enrich the knowledge about the pharmacological treatment of the depressive phases in BD. A series of limitations might influence the interpretation of our results. First, we did not perform a quantitative synthesis with meta-analytical methods. This could impact the validity of our recommendations. However, it should be noted that most studies used heterogeneous measures of outcome, making meta-analytical approaches unfeasible. Second, most of the evidence had low scores for data quality, given the observational nature of the included studies. However, it should be noted that this type of naturalistic analysis has relevance for real-world clinical practice. Third, the studies were too heterogenous to derive recommendations for diagnostic subgroups (BD1 versus BD2). Finally, several factors other than treatment might have influenced the antidepressant outcome (including illness duration, concomitant therapies, age, and psychiatric and somatic comorbidities as well as patients’ experiences and preferences), which should be considered in clinical practice. 

Importantly, some of the identified studies focused on patients with treatment resistance and found positive effects of a ketamine or lamotrigine add-on. Indeed, subsyndromal persisting depressive symptoms are frequent in BD, resulting in low global well-being and psychosocial as well as low cognitive functioning [110]. A pharmacological approach that includes the addition of a compound of proven efficacy to the maintenance treatment could be useful in the clinical management of bipolar depression.

## 4. Materials and Methods

### 4.1. Systematic Search Strategy

We conducted a systematic search (finished on the 5th of April 2022) for peer-reviewed articles in two databases, Medline (via PubMed) and Embase, without any temporal restrictions. The search strategy using Medical Subject Headings (MeSH) in Medline was defined as followed: (“bipolar disorder” [All Fields] AND “depression” [All Fields]) AND (“treatment guidelines” [All Fields] OR “treatment algorithms” [All Fields] OR “drug” [All Fields] OR “treatment” [All Fields] OR “psychotropic” [All Fields]) AND (“humans” [MeSH Terms] AND English [lang]). In Embase, we used the same keywords (bipolar disorder AND depression AND (treatment guidelines OR treatment algorithms OR drug OR treatment OR psychotropic)) limited to keywords in the abstract and English language, humans, articles, Embase status, and journal source. Inclusion criteria required (1) adult humans with the diagnosis of BD with depressive symptoms assessed by the Diagnostic and Statistical Manual of Mental Disorders (DSM) or the International Classification of Diseases (ICD) criteria (all versions); (2) a sample size larger than 20; (3) an outcome of clinical relevance (score scales or clinical global impression); and (4) an observational design, including retrospective or longitudinal prospective studies. Case reports and conference papers were excluded. The search strategy was defined by authors M.M., F.F., and B.C.

### 4.2. Data Management and Selection Process

All results were downloaded from the databases and input into the Rayyan web app [111]. This software consists of an online platform that facilitates the systematic review process, by allowing for sharing among co-authors of the exclusion–inclusion decision of the potentially eligible studies. Data extraction and screening were undertaken by E.C. and F.F., while M.M. checked for accuracy. A PRISMA flowchart [112] describes the screening procedure for the retrieved records (Figure 1). All publications possibly fulfilling eligibility criteria were retrieved for review of the manuscript. The reviews of all manuscripts were conducted by either E.C. and M.M. or F.F. and M.M.

### 4.3. Data Collection Process, Outcomes, and Prioritization

Data were collected through review of the included studies. Initial description of data was focused on reported outcomes. The certainty in the body of evidence was evaluated according to the GRADE approach [113], with individual studies also being initially evaluated according to STROBE [114], with a secondary evaluation by the study’s authors. GRADE provides a structured way of rating quality by evaluating the risk of bias, inconsistency (or heterogeneity) of the results, indirectness, and imprecision. This approach characterizes the certainty of evidence as very low, low, moderate, or high. Due to the naturalistic uncontrolled design of the studies included in this review, only low or very low ratings were possible.

## 5. Conclusions

The findings of observational studies support the recommendations regarding the effectiveness of lamotrigine and quetiapine for bipolar depression. In contrast to the current recommendations, aripiprazole was shown to be effective and generally well-tolerated (except for the onset of akathisia), pointing to it as a valid therapeutic option. Additionally, SSRI were shown to be effective; however, since they were associated with a possibly higher risk of switch to the opposite polarity, they should be used as adjunctive therapy to mood stabilizers. Lithium was shown to be effective, although serum concentrations levels were not associated with clinical response. Finally, ketamine showed a divergent response rate, with so-far unclear long-term effects. In conclusion, this systematic review attempted to summarize the evidence from observational studies.

## Figures and Tables

**Figure 1 pharmaceuticals-16-00182-f001:**
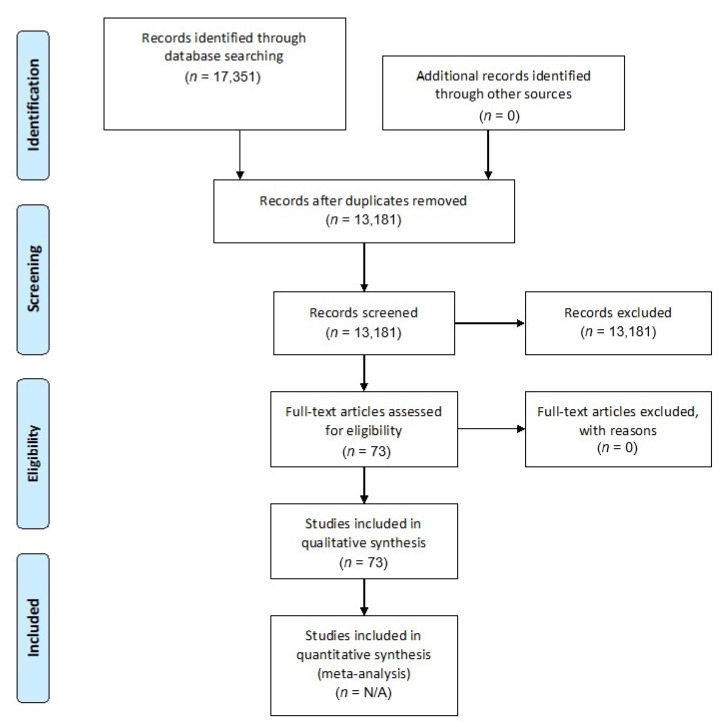
Preferred Reporting Items for Systematic Reviews and Meta-Analyses (PRISMA) flowchart for our systematic review.

**Table 1 pharmaceuticals-16-00182-t001:** Observational studies on anticonvulsants in the treatment of bipolar depression.

Author, Year	Sample	Study Design	Substance	Outcome	Findings	GRADE	Side Effects
Ghaemi et al., 2006 [14]	BD1, BD2, BD NOS, depressed (*n* = 20)	Prospective open label; 8 weeks	Zonisamide	MADRS, SADS	Improvement in all the scales	Very low (50% of patients discontinued; imprecision)	Nausea, vomiting, cognitive impairment, sedation, suicidal ideation, hypomania
McElroy et al., 2005 [15]	BD1 (*n* = 42), BD2 (*n* = 16), BD NOS (*n* = 2), schizoaffective (*n* = 2), all depressive state (*n* = 21)	Prospective open label; 8 weeks	Zonisamide adjunctive therapy	CGI-BP, IDS, YMRS	Depressive patients: decrease in CGI-BP-D and IDS over 8 weeks; 32% classified as responders, with no change in CGI-BP-M or YMRS	Very low (only 7 completed the acute trial; imprecision)	Sedation, tiredness, cognitive impairment, dry mouth, tremors, nausea, diarrhoea, constipation, unsteady gait
Wang et al., 2002 [16]	BD1, BD2, depressed (*n* = 22)	Prospective open label; 12 weeks	Gabapentin	HAMD, YMRS, CGI	Overall HAMD decreased 53%	Low	Mild sedation, weight gain, impaired cognition, hypomanic symptoms (but no change in YMRS)
Wang et al., 2010 [17]	BD2, depressed (*n* = 28)	Prospective open label, uncontrolled; 7 weeks	Divalproex ER	HAMD, MADRS, YMRS, CGI	Improvement in all scales	Low	Weight gain
Dilsaver et al., 1996 [18]	BD, depressed (*n* = 36)	Prospective open label; 3 weeks	Carbamazepine	HAMD	63% remission	Low	Fever and rush
McIntyre et al., 2005 [19]	BD1, BD2; depressed (*n* = 56), mixed (*n* = 42), hypomanic (*n* = 8), manic (*n* = 3)	Prospective open label; 16 weeks	Topiramate	MADRS, YMRS, CGI	Reduction in MADRS and CGI at week 2 and again at week 16; 34% of primarily depressed and 45% of mixed remitted at week 16	Low	Weight loss and reduction in preexisting tremors
Montes et al., 2005 [20]	BD1, 2, NOS, cyclothymic disorder (*n* = 34); start of antidepressant drug (*n* = 28)	Retrospective chart review	Lamotrigine, either monotherapy or add-on	CGI-BP, number of relapses	Reduction in CGI-BP-D (greater in spectrum group than in BD1); 41% relapsed on a depressive episode	Very low (inconsistency/heterogeneity)	Rash, headache. insomnia, dizziness
Ahn et al., 2011 [21]	BD1 (*n* = 15), BD2 (*n* = 22), BD NOS (*n* = 1), depressed	Prospective open label naturalistic; 12 weeks	Add-on of lamotrigine or quetiapine	CGI, GAF	Improvement in both groups. Even better improvement with adjunctive quetiapine	Very low (imprecision)	Dry mouth
Chang et al., 2010 [22]	BD2 (*n* = 109)	Prospective open label naturalistic; 52 weeks	Lamotrigine	CGI-BP-D	Improvement in CGI-BP-S, maximum effect within 12 weeks 64% responded, amongst them 22.9% discontinuation; amongst non-responders 87.2% discontinuation at 12 weeks	Low	12.8% headache, 8.3% dizziness, 6.4% non-serious rash, 5.5% tremors
Joe et al., 2009 [23]	BD depressed (*n* = 259)	Prospective open label naturalistic; 12 weeks	Lamotrigine	CGI-S, development of rashes	Increase response rate in standard titration versus slower titration; not significant in 7th week	Low	Headache, rashes; reduction in rashes in slower titration
Kagawa et al., 2017 [24]	BD1 (*n* = 6), BD2 (*n* = 22), MDD (*n* = 19) depressed	Prospective open label; 8 weeks	Lamotrigine, either monotherapy (*n* = 26) or add-on to valproate (*n* = 20)	MADRS	37% responded, no difference between BD and MDD	Very low	Not reported
Kusumakar et al., 1997 [25]	BD depressed (*n* = 22)	Prospective open label naturalistic; 6 weeks	Lamotrigine	HAMD	72% responders (week 4); 63% remitters (week 6)	Low	Headache, tremors of hands, somnolence, dizziness
Muck-Seler et al., 2008 [26]	BD1 depressed, female (*n* = 26)	Prospective open label; 6 weeks	Lamotrigine	HAMD, YMRS, reduction in MAO-B activity in platelets and clinical correlation	Decrease in HAMD; decrease in platelet MAO-B activity; no correlation between these two factors	Low	Rashes, no serious adverse effects
Sajatovic et al., 2011 [27]	BD1, BD2 (*n* = 57)	Prospective open label; 12 weeks	Lamotrigine	MADRS, HAMD, CGI-BP, WHO-DAS II, UKU	Improvement in MADRS, HAMD, CGI-BP, and most domains of WHO-DAS	Low	None
Sharma et al., 2008 [28]	BD depressed (*n* = 31)	Retrospective chart review	Lamotrigine	CGI-S, CGI-I	Improvement in CGI	Very low	Reduced/increased sleep, weight loss/gain, increased dream activity, polyuria, diminished sexual desire, fatigue
Silveira et al., 2013 [29]	BD1 (*n* = 17), BD2 (*n* = 3) depressed	Prospective pen label naturalistic; 8 weeks	Lamotrigine	HAMD, YMRS, CGI-BD	Improvement in HAMD and CGI; 55% remitted	Very low (imprecision)	25% increase in YMRS (indicating switch to (hypo)mania)

BD = bipolar disorder; BD NOS = bipolar disorder not otherwise specified; CGI = Clinical Global Impression Scale; CGI-BD-D = Clinical Global Impression Scale, bipolar disorder, depression severity; CGI = Clinical Global Impression Scale, bipolar disorder, mania severity; CGI-BP-S = Clinical Global Impression Scale, bipolar disorder, severity; GAF = global assessment of functioning; HAMD = Hamilton Depression Rating Scale; IDS = inventory of depression symptomatology; MADRS = Montgomery Asberg Depression Rating Scale; MDD = major depressive disorder; remission = MADRS ≤ 12; response = ≥ 50% MADRS reduction; SADS = Schedule for Affective Disorders and Schizophrenia; UKU = Udvalg for Kliniske Undersogelser Side-Effect Rating Scale; WHO-DAS = World Health Organization Disability Assessment Schedule; YMRS = Young Mania Rating Scale.

**Table 2 pharmaceuticals-16-00182-t002:** Observational studies on antipsychotics in the treatment of bipolar depression.

Author, Year	Sample	Study Design	Substance	Outcome	Findings	GRADE	**Side Effects**
Bobo et al., 2010 [30]	BD1, BD2, depressed (*n* = 20)	Prospective open label; 8 weeks	Olanzapine	MADRS, QUIDS-SR, CGI	9 positive responses (8 with symptom remission)	Low	Weight gain and somnolence
McInytre et al., 2004 [31]	BD1, BD2, in any state (*n* = 21)	Prospective open label; 6 months	Either risperidone or olanzapine	MADRS YMRS, CGI, AIMS	Reduction in MADRS (risperidone: 17 to 5, olanzapine: 18 to 7); no differences between the groups	Low	Dizziness, somnolence, weight gain
Ahn et al., 2011 [21]	BD1 (*n* = 15), BD2 (*n* = 22), NOS (*n* = 1), depressed	Prospective open label naturalistic; 12 weeks	Add-on of either lamotrigine or quetiapine	CGI, GAF	Improvement in both groups; even better in adjunctive quetiapine	Very low (imprecision)	Dry mouth
Dell’Osso et al., 2012 [32]	MDD (*n* = 10), BD (*n* = 20), with residual depressive symptoms	Prospective, open label naturalistic; 6 weeks	Switch from quetiapine IR to ER	HAMD, HAMA, YMRS, CGI-S, BARS, SDS, compliance, functional impairment	Good efficacy with reduction HAMD and HAMA; 72% scored 100% on BARS; no change in compliance, no change in life quality	Low	Insomnia, drowsiness, weight gain, asthenia, constipation
Kishi et al., 2019 [33]	BD, depressed (olanzapine *n* = 343; quetiapine *n* = 224)	Comparison of two RCTs; 6 weeks olanzapine, 8 weeks quetiapine	Comparison of olanzapine and quetiapine ER	MADRS, HAMD	No difference in response and remission	Moderate	Olanzapine greater risk of weight gain and decreased HDL; quetiapine greater risk of somnolence than olanzapine
Porcelli et al., 2014 [34]	BD, most recent episode depressive (*n* = 21)	Prospective open label; 1 week	Quetiapine ER	HAMD, HAMA	Reduction in HAMD and HAMA total scores; improvement after 3 days	Low	Activity sleepiness, dry mouth, constipation, lack of appetite, tremors, headache, hypotension
Shajahan et al., 2010 [35]	Psychotic and non-psychotic depressed (*n* = 303); thereof BD (*n* = 38)	Retrospective chart review	Quetiapine	CGI-S, CGI-I	69% improvement for all depressive subtypes, best improvement in BD mania, followed by BD depression	Low	Sedation, headache, weight gain, abnormal movements, seizure, gastrointestinal disturbance, low white cell count, paresthesia, prolonged Qtc, blurred vision, edema, sexual dysfunction
Suppes et al., 2007 [36]	BD, depressed (*n* = 19), cycling (*n* = 36)	Retrospective chart review, with prospective rating	Quetiapine adjunctive in acute symptomatology	Life chart	Improvement in depression by week 10 No group difference	Very low	Not reported
Dunn et al., 2008 [37]	BD1 (*n* = 10), BD2 (*n* = 7), BD NOS (*n* = 3), depressed	Prospective open label; Six weeks	Aripiprazole monotherapy and adjunctive treatment	MADRS, YMRS	MADRS improvement, YMRS improved in non-rapid cycling patients	Very low (heterogeneity)	Akathisia, nausea, restlessness, increase in glucose levels (but within normal range)
Kelly et al., 2017 [38]	BD2, BD NOS (*n* = 211)	Retrospective chart review	Aripiprazole	CGI-I, GAF	Improvement in CGI-I and GAF; 21% drop-out because of side events	Low	Akathisia, concentration difficulties, nausea, dizziness, tremor, pain, diarrhea, insomnia, hyperglycemia
Ketter et al., 2006 [39]	BD1, BD2, BD NOS, depressed (*n* = 30)	Prospective open label; No temporal restriction	Aripiprazole	CGI-S, GAF, CMF depressed mood, suicidal ideation score	27% responders, 13% remitters, 47% discontinued (17% inefficacy, 10% patient choice, 20% adverse events)	Very low (heterogeneity, imprecision)	Switch to hypomania, weight change, sedation, nausea, constipation, agitation, cognitive problems; one required cholecystectomy
Malempati et al., 2015 [40]	BD1, BD2, depressed (*n* = 40)	Prospective open label; 2 years	Aripiprazole adjunctive treatment	MADRS, CGI-S, CGI-I, YMRS, SDS	Improvements in MADRS by 6 weeks and CGI-I by six months; complete functional recovery on the Sheehan Disability Scale	Low	Mild weight gain, activation, EPS
Mazza et al., 2008 [41]	BD1, BD2, BD NOS, depressed (*n* = 85)	Prospective open label; 16 weeks	Aripiprazole, either monotherapy or adjunctive	MADRS, CGI-S, YMRS	94.1% decrease in MADRS and CGI scores, regardless of whether monotherapy (22/39 responded, 12/39 remitted) or adjunctive treatment (30/46 responded, 18/46 remitted)	Low	Insomnia, headaches, dizziness, akathisia
Mazza et al., 2009 [42]	BD1 (*n* = 50), depressed	Prospective open label; 16 weeks	Aripiprazole monotherapy	MADRS, SHAPS	Reduction in MADRS: 66% response, 34% remission; Reduction in SHAPS: 52% anhedonia at baseline, 20% at end	Low	Akathisia, headache
McElroy et al., 2007 [43]	BD1, BD2, BD NOS, depressed (*n* = 31)	Prospective open label; 8 weeks	Aripiprazole monotherapy and adjunctive	MADRS, CGI-BP-D, YMRS	Globally 42% responders, 35% remitters; amongst those who completed: 38.5% responders, 30.8% remitters (monotherapy); 44.4% responders, 38.9% remitters (adjunctive therapy)	Low	29% discontinuation because of side effects: akathisia, insomnia, activation, nausea, increased/decreased appetite, headache, tremor, anxiety, concentration difficulty, fatigue, blurred vision, increased urinary frequency, muscle soreness, manic symptoms
Ketter et al., 2016 [44]	BD1 (*n* = 817)	Prospective open label multicenter; 24 weeks	Lurasidone monotherapy (38.9%) and adjunctive to lithium or valproate (61.1%)	MADRS, CGI-BP-S, HAMA, Q-LES-Q-SF, SDS, BARS, AIMS, YMRS, C-SSRS	Improvement from baseline study in all scales	Moderate	6.9% in monotherapy and 9% of adjunctive group discontinuation because of adverse effects; during extension period: worsening depression, suicidal ideation, bone fractures, suicide attempt, mania
Miller et al., 2018 [45]	BD1 (*n* = 32), BD2 (*n* = 26), NOS (*n* = 3)	Prospective open label naturalistic; no temporal restriction	Lurasidone, mainly adjunctive	Discontinuation, CGI-BP-S	CGI-BP-S decreased in depressed (5.2 to 4.3); no change observed in subsyndromal depression	Low	54% discontinued because of side effects: Akathisia, sedation, weight gain
Liebowitz et al., 2009 [46]	BD2, depressed (*n* = 30)	Prospective open label; 8 weeks	Ziprasidone	HAMD, HAMA, MADRS, YMRS, CGI-S, BDI, Q-LES-Q	30% responders and 17% remitters (after 1 week), 60% responders, 43% remitters by end of the treatment	Low	Muscle stiffness, insomnia, low white blood cell count, nervousness, tremor, headache, mood swings, drowsiness
Mac Fadden et al., 2011 [47]	BD1, BD2, depressed (*n* = 59), manic/mixed (*n* = 103)	Prospective open label first phase; 12 weeks	LAI risperidone	CGI-BP-S, MADRS, YMRS	53.3% remission	Low	Tremor, muscle rigidity, weight increase, headache
Brown et al., 2019 [48]	BD1, BD2, most recent episode depressed (*n* = 21)	Prospective open label; 8 weeks	Brexpiprazole	MADRS, IDS-SR30, QOLBD, RAVLT, TMT	Reduction in MADRS and IDS-SR30	Low	Akathisia

AIMS = Abnormal Involuntary Movement Scale; BARS = Barnes Akathisia Rating Scale; BD = bipolar disorder; BD NOS = bipolar disorder not otherwise specified; CGI = Clinical Global Impression Scale; CGI-BD-D = Clinical Global Impression Scale, bipolar disorder, depression severity; CMF = Clinical Monitoring Form; C-SSRS = Columbia Suicide Severity Rating Scale; BDI = Becks’ Depression Inventory; CGI = Clinical Global Impression Scale, bipolar disorder, mania severity; ER = extended release; GAF = Global Assessment of Functioning; HAMD = Hamilton Depression Rating Scale; HAMA = Hamilton Anxiety Rating Scale; HDL = high-density lipoprotein; IDS = Inventory of Depression Symptomatology; IR = immediate release; LAI = long-acting injectable; MADRS = Montgomery Asberg Depression Rating Scale; MDD = major depressive disorder; Q-LES-Q-SF = Quality of Life, Enjoyment, and Satisfaction Questionnaire—Short Form; QOLBD = quality of life in bipolar disorder; QUIDS-SR = Quick Inventory Of Depressive Symptomatology (Self Report); RAVLT = Rey Auditory Verbal Learning Test; RCT = randomized control trial; remission = MADRS ≤ 12; response = ≥ 50% MADRS reduction; SADS = Schedule for Affective Disorders and Schizophrenia; SDS = Sheehan Disability Scale; SHAPS = Snaith–Hamilton Pleasure Scale; TMT = Trail Making Test; UKU = Udvalg for Kliniske Undersogelser Side-Effect Rating Scale; WHO-DAS = World Health Organization Disability Assessment Schedule; YMRS = Young Mania Rating Scale.

**Table 3 pharmaceuticals-16-00182-t003:** Observational studies on antidepressants in the treatment of bipolar depression.

Author, Year	Sample	Study Design	Substance	Outcome	Findings	GRADE	**Side Effects**
Amsterdam et al., 1997 [56]	BD, depressed (*n* = 49), MDD, depressed (*n* = 566)	Prospective open label, 8 weeks	Fluoxetine (20 mg)	HAMD	411 responders; no correlation between plasma concentration and clinical outcome	Low	Not reported
Amsterdam et al., 2010 [57]	BD, depressed (*n* = 148)	Prospective open label; 14 weeks	Fluoxetine monotherapy (20–80 mg)	HAMD, YMRS	88 responders, 86 remitters; mean time to remission 64.4 days	Low	*n* = 6 hypomania headache, yawning, nausea, reduced appetite, upper respiratory tract infection, decreased libido, delayed orgasm, increased blood pressure
Amsterdam et al., 2013 [58]	BD, rapid cycling (*n* = 42), BD, non-rapid cycling (*n* = 124); same cohort [57]	Prospective open label; 14 weeks	Fluoxetine monotherapy (10–80 mg)	HAMD, YMRS, CGI	Response and remission comparable in both groups; higher decrease in HAMD score in rapid cycling	Low	*n* = 6 hypomania, equal in both groups, *n* = 1 attempted suicide, *n* = 1 manic episode
Baldassano et al., 1995 [59]	BD1 (*n* = 19), BD2 (*n* = 1)	Retrospective chart review	Paroxetine (10–40 mg), mostly adjunctive	HAMD, CGI, GAF	65% improved “much” or “very much”; GAF mean improved from 44.4 to 60.4	Very low (heterogeneity)	*n* = 1 hypomania, *n* = 1 rapid cycling, both with history of drug-induced switch
Fonseca et al., 2006 [60]	BD1, BD2, depressed (*n* = 20)	Prospective open label; 12 weeks	Escitalopram (10 mg), adjunctive	HAMD, CGI-S, CGI-I, YMRS	Decrease in HAMD (mean 20.9 baseline versus 8.9 end), CGI-S (4.8 versus 1.5), CGI-I	Low	*n* = 4 discontinuation, because of switch (*n* = 1), hypomanic symptoms (*n* = 2), hospitalization, psychosis, suicidal ideation (*n* = 1); 75% had at least one adverse effect: headache, somnolence, nausea, mood switch, suicidal ideation
Kupfer et al., 2001 [61]	BD1, BD2 (*n* = 45; 12 dropped out before week 8)	Prospective open label; 8 weeks	Citalopram (20–80 mg), adjunctive	HAMD, CGI-I, YMRS	21/33 responded in all scales after 8 weeks; of them, 14/19 remitted after 16 additional weeks	Low	37/45 only mild to moderate headache, nausea, diarrhea, sexual dysfunction, mania; no study discontinuation because of adverse events
Mertens et al., 1989 [62]	BD or MDD, depressed (*n* = 70)	Prospective double blind; 6 weeks	Paroxetine (30 mg) or mianserin (60 mg)	HAMD	Improve in HAMD in paroxetine (28.5 baseline versus 11.5 end) and mianserin (30.8 versus 17.8); no group difference	Low	Both drugs well-tolerated, with nausea and headache in four patients, with somnolence in six patients
Serafini et al., 2010 [63]	BD (*n* = 49), MDD (*n* = 13)	Prospective open label; 12 weeks	Duloxetine or venlafaxine, mostly adjunctive	HAMA, HAMD, SF-36	Duloxetine (90.3% response, 48.4% remission with < 8) more effective in all scales than venlafaxine	Low	Hypertension (with venlafaxine), nausea (duloxetine), hypomania (both drugs)
Kocsis et al., 1990 [64]	Psychotic depressed: BD (*n* = 12), MDD (*n* = 13); severely depressed: BD (*n* = 13), MDD (*n* = 40); moderately depressed: BD (*n* = 22), MDD (*n* = 32)	Prospective open label; 4 weeks	Amitryptiline or imipramine for four weeks, after 2 weeks drug-free (placebo)	HAMD, SADS, depression severity	Good outcome in 67% of moderately (HAMD 21 baseline versus 10 end) versus 39% severely depressed (33 versus 17) versus 32% psychotic depressed (35 versus 22) Better response in moderate than severe depression; no difference between severly depressed with and without psychosis; no differences between BD and MDD	Low	Not reported
Calabrese et al., 2007 [65]	BD1, depressed (*n* = 21)	Prospective open label; 6 weeks, optional +46 weeks	Agomelatine (25 mg), adjunctive to lithium or valproate	HAMD, MADRS, CGI	81% response; 47.6% response after 1 week; no difference in taking lithium or valproate	Low	Anxiety, agitation, breast abscess, social problem, bereavement reaction, traffic accident; one drug-related manic switch
Fornaro et al., 2013 [66]	BD2, depressed (*n* = 28)	Prospective open label; 6 weeks, optional +30 weeks	Agomelatine (25 mg), adjunctive to lithium or valproate	HAMD, YMRS	64% response after six weeks and 86% after 36 weeks; taking lithium responded in 54.5% after 6 weeks and in 90.9% after 36 weeks and taking valproate in 70.6 respectively 82.4%	Low	Four patients with valproate and agomelatine: pseudo-vertigo and hypomania; two patients with lithium and agomelatine: insomnia and mania and, therefore, dropped out at week 6; Two more cases of hypomania at week 36
Ionescu et al., 2015 [67]	BD1, BD2 (*n* = 36)	Prospective open label; spin-off of double-blind RCT; single infusion	Ketamine 0.5 mg/kg IV as adjunctive therapy to lithium or valproate	MADRS, HDRS, HAM-A, CADSS	Significant reduction in all scales, in both anxious (MADRS 33 to 18) and non-anxious groups (MADRS 33 to 20)	Low	Unspecified; no difference between anxious and non-anxious groups
Lara et al., 2013 [95]	BD, MDD (*n* = 26)	Prospective open label; 1–20 doses every 2–3 days	Ketamine sublingual, start 0.1 mL, up titration up to 10 mg, then 7 stable doses	Not standardized questions about mood/sleep/cognition	77% remission or clear response on depression, mood instability, cognitive impairment, poor sleep	Very low (imprecision)	No manic, psychotic, or dissociative symptoms were observed, but two bipolar patients reported agitation for a few hours; mild light-headedness was a common but transient side-effect, subsiding typically in < 30 min, more pronounced or present only after the first dose
Li et al., 2022 [76]	BD, MDD (*n* = 109)	Prospective open label; 6 infusions in 12 days; 9-month observation	Ketamine 0.5 mg/kg IV	PHQ-9, GAF	Of 56 responders, 46.4% remained stable after 9 months, 25% relapsed within two weeks	Low	Not reported
McIntyre et al., 2020 [68]	BD (*n* = 30), MDD (*n* = 183); After 4 infusions, *n* = 107	Prospective open label; 4 infusions in 7–8 days	Ketamine 0.5 mg/kg IV as adjunctive	QUIDS, GAD-7, SDS	27% response (QUIDS total score reduction ≥ 50%, 13% remission; positive effect on anxiety, overall psychosocial function, suicidal ideation	Low	During/after infusion: 48.2%/49.2% dizziness, 57.1%/53.1% drowsiness, 43.5%/25.1% confusion, 38.2%/17.6% depersonalization, 40.8%/16.7% derealization, 31.9%/29.8% blurred vision, 20.8%/18.8% double vision, 13.9%/11.1% nausea, 13.2%/19.3% headache
McIntyre et al., 2021 [69]	BD (*n* = 48), MDD (*n* = 259, other depressed (*n* = 11), after 4 infusions, *n* = 142); partly overlapping cohort [68]	Prospective open label; 4 infusions in 7–8 days	Ketamine 0.5 mg/kg IV as adjunctive	QUIDS, SDS	Total functional disability; the subdomains social life and family life/home responsibilities significantly moderate ketamine response	Low	Not reported
Pennybaker et al., 2018 [70]	BD, MDD (*n* = 122)	Data of four open, prospective, partially placebo-controlled trials; single infusion	Ketamine 0.5 mg/kg IV, in BD adjunctive to lithium or valproate	MADRS, HAMD, YMRS, CADDS, BDI	32.5% antidepressant response after 24 h, 12.9% after two weeks (only 93/122 were assessed at two weeks);responders at week two had a greater response after 230 min and after 24 h than two-week non-responders	Low	Not reported
Permoda-Osip et al., 2013 [71]	BD1, BD2 (*n* = 20)	Prospective open label; single infusion	Ketamine 0.5 mg/kg IV, adjunctive	HAMD	10/20 responders	Low	Not reported
Rybakowski et al., 2013 [72]	BD, depressed (*n* = 25)	Prospective open label; single infusion	Ketamine 0.5 mg/kg IV	HAMD	After 24 h: 6/20 responders, 4/20 remitters; after 7 days: 13/20 responders, 8/20 remitters; no correlation with neurotrophins (apart from reduction in BDNF levels after 7 days in non-responders)	Low	Not reported
Zheng et al., 2018 [73]	BD, depressed (*n* = 20); MDD, depressed (*n* = 77)	Prospective open label; 6 infusions in 12 days	Ketamine 0.5 mg/kg IV, adjunctive	MADRS, SSI, HAM-A, BPRS, CADSS, HAMD	Response rate 68%, remission rate 50.5%	Low	Mild temporary dissociative and psychotomimetic symptoms. No differences between responders and non-responders

AIMS = Abnormal Involuntary Movement Scale; BPRS = Brief Psychiatric Rating Scale; BD = bipolar disorder; BD NOS = bipolar disorder not otherwise specified; CGI = Clinical Global Impression Scale; CGI-BD-D = Clinical Global Impression Scale, bipolar disorder, depression severity; CMF = Clinical Monitoring Form; BDI = Becks’ Depression Inventory; CGI = Clinical Global Impression Scale, bipolar disorder, mania severity; ER = extended release; GAF = Global Assessment of Functioning; HAMD = Hamilton Depression Rating Scale; HAMA = Hamilton Anxiety Rating Scale; IDS = Inventory of Depression Symptomatology; IR = immediate release; LAI = long-acting injectable; MADRS = Montgomery Asberg Depression Rating Scale; MDD = major depressive disorder; Q-LES-Q = Quality of Life, Enjoyment, and Satisfaction Questionnaire; QOLBD = quality of life in bipolar disorder; QUIDS-SR = Quick Inventory Of Depressive Symptomatology (Self Report); RAVLT = Rey Auditory Verbal Learning Test; RCT = randomized control trial: remission (if not differently described) = MADRS ≤ 12; response (if not differently described) = ≥ 50% MADRS/HAMD reduction; SADS = Schedule for Affective Disorders and Schizophrenia; SDS = Sheehan Disability Scale; SF36 = Short Form Survey; SHAPS = Snaith–Hamilton Pleasure Scale; TMT = Trail Making Test; UKU = Udvalg for Kliniske Undersogelser Side-Effect Rating Scale; WHO-DAS = World Health Organization Disability Assessment Schedule; YMRS = Young Mania Rating Scale.

## Data Availability

Data sharing not applicable.

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
