# Peer review of "Pharmacological Treatment of Bipolar Depression: A Review of Observational Studies"

_pharmaceuticals, 2023, doi:10.3390/ph16020182_

Round 1

Reviewer 1 Report

This submission reports a thorough systematic review addressing an important topic, that is, the treatment of bipolar depression in real-world settings, providing a comprehensive overview of the available evidence and grading its quality. This methodologically sound article is very relevant to both researchers and end-users such as clinicians and policymakers so to pinpoint which options are the best ones to deal with an often difficult-to-treat condition.

The article has no major flaws, but I have one suggestion for the Authors that can hopefully further increase the significance of their discussion.

At lines 556-560, the Authors quickly hint at possible alternative options for the management of bipolar depression, but they only give a partial view of the matter. Since I believe that this topic deserves greater insight, I would suggest that the Authors refer to a recent, relevant umbrella review comprehensively synthesizing and grading the existing evidence regarding drug repurposing for bipolar depression [Bartoli et al., 2021, https://doi.org/10.1016/j.jpsychires.2021.09.018].

Author Response

This submission reports a thorough systematic review addressing an important topic, that is, the treatment of bipolar depression in real-world settings, providing a comprehensive overview of the available evidence and grading its quality. This methodologically sound article is very relevant to both researchers and end-users such as clinicians and policymakers so to pinpoint which options are the best ones to deal with an often difficult-to-treat condition.

R) We thank the reviewer for its positive assessment of our work.

Q1) The article has no major flaws, but I have one suggestion for the Authors that can hopefully further increase the significance of their discussion. At lines 556-560, the Authors quickly hint at possible alternative options for the management of bipolar depression, but they only give a partial view of the matter. Since I believe that this topic deserves greater insight, I would suggest that the Authors refer to a recent, relevant umbrella review comprehensively synthesizing and grading the existing evidence regarding drug repurposing for bipolar depression [Bartoli et al., 2021, https://doi.org/10.1016/j.jpsychires.2021.09.018].

R1) We thank the reviewer for the positive evaluation of our work. The suggested reference was added to the Discussion: “Future directions for clinical research should lead to the identification of novel drugs for the management of this difficult to treat mood phase. This might be achieved through pre-clinical research and development brand new molecules which could be tested in properly designed trials, or alternatively through the repurposing of drugs indicated for other conditions [115]. Indeed, a recent meta-review of Bartoli and co-authors showed that repurposed drugs such as modafinil/armodafinil, pramipexole, celecoxib and N-acetylcysteine demonstrated varying degree of efficacy in bipolar depression although generally the quality of evidence was low.

Reviewer 2 Report

The comprehensive review of the naturalistic evidence. Content is fine, but orgnization of the manuscript is a bit odd: Why did you place the methodology section between Discussion und Conclusions, and not in front of Resullts? I suggest to follow the usual flow

Author Response

Q1) The comprehensive review of the naturalistic evidence. Content is fine, but orgnization of the manuscript is a bit odd: Why did you place the methodology section between Discussion und Conclusions, and not in front of Results? I suggest to follow the usual flow

R1) Thank you for this observation. The sequence of the main sections follows the formatting guidelines of the Journal.

Reviewer 3 Report

This is a very important paper not only from a scientific aspect but also with direct clinical implication. The topic chosen is of high importance, the introduction is comprehensive yet concise, the aims are relevant and clearly stated, the applied methodology is sufficient to respond to the study questions. As for the methods, the authors strictly followed requirements for systematic reviews and rigorously identified and selected the relevant papers, and summarised their information in a reliabble, intelligible and once again clinically relevant way. The discussion of the findings strictly follow the results and is very well structured. The authors draw only fully supported conclusions.

As the paper is methodologically sound, well written, and important I believe it can be accepted for publication in its present form and no corrections are necessary. Congratulations to the authors.

Author Response

This is a very important paper not only from a scientific aspect but also with direct clinical implication. The topic chosen is of high importance, the introduction is comprehensive yet concise, the aims are relevant and clearly stated, the applied methodology is sufficient to respond to the study questions. As for the methods, the authors strictly followed requirements for systematic reviews and rigorously identified and selected the relevant papers, and summarised their information in a reliable, intelligible and once again clinically relevant way. The discussion of the findings strictly follow the results and is very well structured. The authors draw only fully supported conclusions. As the paper is methodologically sound, well written, and important I believe it can be accepted for publication in its present form and no corrections are necessary. Congratulations to the authors.

R1) We are very grateful for the positive comments provided by the reviewer.

Reviewer 4 Report

The review manuscript deals with a crucial topic of interest - pharmacotherpay of bipolar depression. 

Introduction is well drafted. The authors can check the order of sections. Whether methods and search strategies come after results or before?

The results were well tabulated. 

Discussion was within the limits of obtained results. The authors can highlight what future research can be carried out to bridge the literature gap. 

Author Response

Q1) The review manuscript deals with a crucial topic of interest - pharmacotherapy of bipolar depression. Introduction is well drafted. The authors can check the order of sections. Whether methods and search strategies come after results or before?

R1) Thank you for this observation. The sequence of the main sections follows the formatting guidelines of the Journal.

Q2) The results were well tabulated. Discussion was within the limits of obtained results. The authors can highlight what future research can be carried out to bridge the literature gap.

R2) Thank you, a brief section on possible future developments for research has been added to the Discussion: “Future directions for clinical research should lead to the identification of novel drugs for the management of this difficult to treat mood phase. This might be achieved through pre-clinical research and development brand new molecules which could be tested in properly designed trials, or alternatively through the repurposing of drugs indicated for other conditions [115]. Indeed, a recent meta-review of Bartoli and co-authors showed that repurposed drugs such as modafinil/armodafinil, pramipexole, celecoxib and N-acetylcysteine demonstrated varying degree of efficacy in bipolar depression although generally the quality of evidence was low.”